# Birth Weight and Early Postnatal Outcomes: Association with the Cord Blood Lipidome

**DOI:** 10.3390/nu14183760

**Published:** 2022-09-12

**Authors:** Carolina Gonzalez-Riano, Marcelo Santos, Marta Díaz, Cristina García-Beltran, Carles Lerin, Coral Barbas, Lourdes Ibáñez, David Sánchez-Infantes

**Affiliations:** 1Centro de Metabolómica y Bioanálisis (CEMBIO), Facultad de Farmacia, Universidad San Pablo-CEU, CEU Universities, Urbanización Montepríncipe, 28660 Boadilla del Monte, Spain; 2Endocrinology Department, Institut de Recerca Sant Joan de Déu, 08950 Barcelona, Spain; 3Centro de Investigación Biomédica en Red de Diabetes y Enfermedades Metabólicas Asociadas, Instituto de Salud Carlos III, 28029 Madrid, Spain; 4Centro de Investigación Biomédica en Red de Fisiopatología de la Obesidad y Nutrición (CIBERobn), 28029 Madrid, Spain; 5Department of Health Sciences, Campus Alcorcón, University Rey Juan Carlos (URJC), 28922 Madrid, Spain

**Keywords:** lipidomics, LC–MS, lipid metabolism, gestational age, birth weight, cord blood serum

## Abstract

Being born small or large for gestational age (SGA and LGA, respectively), combined with suboptimal early postnatal outcomes, can entail future metabolic alterations. The exact mechanisms underlying such risks are not fully understood. Lipids are a highly diverse class of molecules that perform multiple structural and metabolic functions. Dysregulation of lipid metabolism underlies the onset and progression of many disorders leading to pathological states. The aim of this pilot study was to investigate the relationships between birth weight, early postnatal outcomes, and cord blood serum lipidomes. We performed a non-targeted lipidomics-based approach to ascertain differences in cord blood lipid species among SGA, LGA, and appropriate-for-GA (AGA) newborns. Moreover, we longitudinally assessed (at birth and at ages of 4 and 12 months) weight and length, body composition (DXA), and clinical parameters. We disclosed distinct cord blood lipidome patterns in SGA, LGA, and AGA newborns; target lipid species distinctly modulated in each SGA, AGA, and LGA individual were associated with parameters related to growth and glucose homeostasis. The distinct lipidome patterns observed in SGA, AGA, and LGA newborns may play a role in adipose tissue remodeling and future metabolic risks. Maternal dietary interventions may potentially provide long-term benefits for the metabolic health of the offspring.

## 1. Introduction

Being born small for gestational age (SGA; birth weight (BW) < −2 SD [1]) or large for GA (LGA; BW > +2 SD [2]) may be linked to metabolic complications later in life, depending on postnatal outcomes [2,3,4,5,6,7,8]. Several studies have demonstrated that SGA infants who experience a rapid and exaggerated postnatal catch-up in weight are at greater risk of developing metabolic abnormalities, including excess ectopic (hepato-visceral) or central fat (due to a limited capacity of subcutaneous adipose tissue for fat storage), insulin resistance, and a thicker carotid intima–media [4,5,7,9,10]. In addition, catch-up girls tend to develop precocious pubarche (i.e., appearance of pubic hair before 8 years of age) due to premature adrenarche, which can be followed by an early and rapidly progressive puberty/menarche, and by greater risks for developing polycystic ovary syndrome in adolescence [11,12]. On the other hand, LGA infants born to mothers without diabetes and without excessive gestational weight gain tend to have more subcutaneous adipose tissue at birth, and become relatively lean early in postnatal life [13]. However, the mechanisms underlying those outcomes are not fully understood. Here, we hypothesized that SGA and LGA newborns would present distinct lipidomic profiles as compared to AGA newborns, and that those lipid classes would be associated with longitudinal clinical and/or endocrine–metabolic parameters.

Lipids are a highly diverse molecular class essential for multiple physiological functions, including energy storage and membrane architecture or signaling [14]. Dysregulation of lipid metabolism plays a major role in the onset and progression of many pathological states. A link between early-gestation maternal lipid levels and cord blood lipidomes and their relationship with BW has been reported [15], with positive associations between BW and branched-chain amino acids and phosphatidylcholines [16,17,18]. Moreover, dynamic shifts in the maternal lipidome over the course of pregnancy, including transfer of lipids containing polyunsaturated fatty acids (PUFAs) from maternal to fetal circulation, have been demonstrated, suggesting that maternal lipids may modulate cord blood lipid levels, influencing growth [15]. In addition, the role of n-6 and n-3 fatty acids and their metabolism into biologically active eicosanoids is still a milestone in the field, since eicosanoids may play a key role in inflammation [19].

The use of lipidomics-based approaches as disease-monitoring and diagnostic tools using ultrahigh-pressure liquid chromatography coupled with mass spectrometry (UHPLC–MS) is well recognized in the field of inborn errors of metabolism [20,21]. Along these lines, the comprehensive lipidomic profiling of cord blood could help to unveil potential biomarkers of postnatal growth and future disease risk. In the present pilot study, we performed a non-targeted lipidomics-based approach using ultrahigh-pressure liquid chromatography—accurate-mass quadrupole time-of-flight mass spectrometry with electrospray ionization (UHPLC-ESI-QTOF MS) in positive and negative ion modes to ascertain potential differences in cord blood lipid species among SGA, LGA, and AGA newborns. Subsequently, we assessed the relationships between BW, postnatal outcomes, and cord blood lipid levels.

## 2. Material and Methods

### 2.1. Study Population, Study Design & Ethics

The study population consisted of 36 mother–infant pairs recruited between 2006 and 2012 at Hospital Sant Joan de Déu in Barcelona, within two prospective longitudinal studies assessing body composition and the endocrine–metabolic state of SGA, AGA, and LGA subjects during infancy [2,22,23]. Inclusion criteria were as follows: (1) uncomplicated, singleton, term (37–42 weeks) pregnancy; (2) delivery of SGA, AGA, or LGA newborns, defined as BW Z-scores below −2 SD, between −1 and +1 SD, and above +2 SD, respectively [2,22,23]; (3) exclusive breast- or formula feeding for at least 4 months; (4) sample availability for performing lipidomic assessment; (5) written informed consent from the pregnant mothers before delivery. Exclusion criteria were maternal hypertension, preeclampsia or diabetes mellitus, alcohol or drug abuse, complications at birth (i.e., need for resuscitation or parenteral nutrition), maternal medications, and evidence of congenital malformations.

Twelve newborns were born SGA (*n* = 6 girls and *n* = 6 boys); 12 were AGA (*n* = 7 girls and *n* = 5 boys), and 12 were LGA (*n* = 6 girls, and *n* = 6 boys). Postnatal follow-up to the age of 12 months was completed in all participants. The study was approved by the Institutional Review Board of Hospital Sant Joan de Déu in Barcelona, with the project identification code ‘PIC-130-18′, entitled ‘Longitudinal profile of circulating exosomes and their proteome in children born with low birth weight and adequate weight for gestational age: usefulness as a biomarker of metabolic syndrome’, dated 28 March 2019. The SGA, AGA, and LGA infants selected for the present study had comparable anthropometric, endocrine–metabolic, and body composition characteristics to those of the remaining newborns within each specific subgroup.

### 2.2. Clinical, Endocrine–Metabolic, and Body Composition Assessments

Maternal age at delivery, smoking habits, parity, pre-gestational weight, and weight and body mass index (BMI) at delivery (the latter defined as the weight (kg)/square of height (m)) were obtained from the hospital clinical records. Gestational age was estimated according to the last menses, and confirmed by first-trimester ultrasound.

Infants’ weight and length were measured immediately after delivery, and again at the ages of 4 and 12 months, from which BMI was derived. Sex- and gestational-age-adjusted Z-scores for anthropometric measurements were calculated using Spanish normative data [24]. Umbilical cord blood samples were collected immediately after birth, and processed before the separation of the placenta, as previously described [25]. Venous samples at the ages of 4 and 12 months were obtained in the fasting state during the morning; the serum fraction was separated by centrifugation and stored at −80 °C until analysis.

Serum glucose was determined by the glucose oxidase method. Serum insulin and insulin-like growth factor-I (IGF-I) were measured by immunochemiluminiscence (DPC IMMULITE 2500, Siemens, Germany); the detection limit for IGF-I was 25 ng/mL; the intra- and inter-assay coefficients of variation (CVs) were <10%. Homeostasis model assessment of insulin resistance (HOMA-IR) was assessed as fasting insulin (mU/L) × fasting glucose (mmol/L)/22.5. Circulating high-molecular-weight (HMW) adiponectin was measured with a specific human ELISA (R&D Systems, Minneapolis, MN); the intra- and inter-assay CVs were <9%.

Body composition was assessed at the age of 15 days, and again at 4 and 12 months, by absorptiometry with a Lunar Prodigy, coupled to Lunar software (version 3.4/3.5; Lunar Corp, Madison, WI, USA), adapted for infants [5]. Body fat, lean mass, and abdominal fat were assessed during natural sleep. CVs were <3% for fat and lean mass.

### 2.3. Lipidomics Fingerprinting by UHPLC-ESI-QTOF MS

Lipid extraction from umbilical cord samples was performed as previously described [26]. Briefly, 40 µL of serum was mixed with 800 µL of methanol: MTBE: chloroform (4:3:3, *v*/*v*/*v*), which contained 1.08 ppm of sphinganine (d17:0) as the internal standard (IS) for ESI(+), and 4.03 ppm of palmitic acid-d_31_ as the IS for ESI(−). Then, 20 µL of 5 ppm LightSPLASH^®^ LIPIDOMIX^®^ primary standard mixture stock solution in MeOH was added [27]. The mixture of lipid internal standards was added for lipid quantification and semi-quantification purposes (see the Appendix A, for more information about ISs and the semi-quantification process). Samples were vortexed thoroughly for 20 min at room temperature, and then centrifuged (16,100× *g*, 5 min, 15 °C) before transferring them into sample vials with glass inserts for LC–MS analysis.

Quality control samples (QCs) were prepared by pooling equal volumes of each serum sample, and were processed identically in parallel with the rest of the study samples. The samples were then randomized, and QCs were injected at the beginning, every five experimental samples, and at the end of the batch. Four blank samples were prepared along with the rest of the samples, following the same lipid extraction procedure. The blank samples were then analyzed at the beginning and the end of the analytical sequence to identify common contaminations. Finally, serum extracts were analyzed using an Agilent 1290 Infinity II UHPLC system coupled to an Agilent 6546 quadrupole time-of-flight (QTOF) mass spectrometer in both positive and negative ion modes, using the analytical conditions previously described [20] (see Appendix A: Supporting information [28,29,30,31,32,33,34,35]).

### 2.4. Data Processing

Data collected after the LC–MS analysis were cleaned from background noises and unrelated ions by recursive analysis using the MassHunter Profinder software (B.10.0.2, Agilent Technologies, Santa Clara, CA, USA). First, the molecular feature extraction (MFE) algorithm was used to perform the chromatographic deconvolution to build all of the mass spectral data features, which were the sum of coeluting ions that were related by a charge-state envelope, isotopologue pattern, and/or the presence of different adducts and dimers in the analyzed samples. In parallel, the MFE aligned the molecular features across the study samples using the mass and RT to build a single spectrum for each compound group. Subsequently, the MFE results were used to perform recursive feature extraction, where the batch find-by-ion extraction (FbI) algorithm uses the median mass, median retention time, and the composite spectrum calculated from the aligned features to improve reliability [28]. To detect coeluting adducts of the same features, the following adducts were selected: [M + H]^+^, [M + Na]^+^, [M + K]^+^, [M + NH_4_]^+^ and [M + C_2_H_6_N_2_ + H]^+^ in LC-ESI(+)-MS; [M − H]^-^, [M + Cl]^−^, [M + CH_3_COOH − H]^−^, and [M + CH_3_COONa − H]^−^ in LC-ESI(−)-MS. The neutral loss (NL) of water was also considered for both ion modes.

### 2.5. Lipidomic Data Normalization and Analysis

Data normalization and filtration were performed prior to statistical analysis. First, the CVs of both ISs were calculated. The raw data matrices obtained were then normalized according to the intensity of the corresponding IS to correct the unwanted variance related to sample preparation and the analytical run. Then, the features were selected based on their CVs in the QCs, and a cutoff threshold of 20% for LC–MS was set for the CV values of lipids present in the QC samples. Features with mean blank values higher than 10% of the sample mean value were considered to be non-relevant [36].

Differences between subgroups were investigated by both univariate (UVDA) and multivariate (MVDA) data analyses. Regarding UVDA, distinctions between the three birth weight groups were evaluated for each lipid using MATLAB (2018, MathWorks, Netik, MA, USA) with the Kruskal–Wallis test (*p* ≤ 0.05) after normality testing with the Shapiro–Wilk test. Post hoc, pairwise analyses were carried out using the Mann–Whitney U test to conclude whether a specific lipid was significant or not in a comparison (LGA vs. AGA; SGA vs. AGA; LGA vs. SGA). Finally, the false discovery rate at the level α = 0.05 was inspected using the Benjamini–Hochberg correction test. Compounds registered with a Kruskal–Wallis *p*-value slightly over 0.05 were retained, enhancing the biological information of the study based on their significant *p*-values obtained in at least one of the comparisons performed with Mann–Whitney U test, and considering that their levels followed the same trend as the lipid species of their class. For MVDA (SIMCA P + 16.0), Pareto scaling and logarithmic transformation were applied before generating the unsupervised principal component analysis (PCA-X), partial least squares discriminant analysis (PLS-DA), and orthogonal partial least squares discriminant analysis (OPLS-DA) models. The tightness of QCs’ clustering displayed in the PCA plots was used to assess the analytical procedures’ reliability and robustness. PLS-DA was then performed to expose the global lipidomic changes due to BW, and the groups were compared using the OPLS-DA model to maximize class discrimination and explore the driving forces among the variables. The variable influence on projection (VIP) values were computed using the OPLS-DA model, keeping those lipids with a VIP ≥ 1 and a jackknife confidence interval value other than zero. A representative cross-validated OPLS-DA model with LC–MS ESI(+) and LC–MS ESI(−) data, including the three subgroups, was generated. Finally, the OPLS-DA models were validated with cross-validation and the CV-ANOVA tool provided by the SIMPA-P+ software.

### 2.6. Lipid Annotation

The annotation workflow [28] consisted of three steps: (i) initial tentative identification of lipid features based on the MS1 data using the online tool CEU Mass Mediator (CMM) (http://ceumass.eps.uspceu.es/mediator/ (accessed on 4 July 2021)) [29,30], (ii) reprocessing of the raw LC–MS/MS data with Lipid Annotator software (Agilent Technologies Inc., Santa Clara, CA, USA) and, finally, (iii) manual MS/MS spectral interpretation using the Agilent MassHunter Qualitative Analysis software (version 10.0), comparing the retention time and MS/MS fragmentation with the available spectral data included in several databases [31,32,33]. Detailed information about the lipid annotation process can be found in the Appendix A. The lipid nomenclature convention used here for the lipid species reported follows the latest update of the shorthand annotation [37]. Once the most affected lipids were annotated, the Lipid Network Explorer (LINEX) [38] tool was used to visualize and analyze the lipidomics networks generated with the specific sets of lipids previously marked as being the most distinctive within the SGA and LGA subgroups when compared to the AGA subgroup, and also the most distinctive sets of lipids within each of the SGA and LGA subgroups for comparisons between them.

### 2.7. Statistical Analysis

Descriptive variables are expressed as the mean ± SEM. We used JMP 16 software (SAS, Cary, NC, USA) to analyze and interpret the statistical data. In the presence of variables with non-normal distribution, data were logarithmically transformed before analysis, and corrected by applying Log+1. The significance level was set at *p* ≤ 0.05. Bivariate correlations were performed to study associations between lipids and maternal and infant variables.

## 3. Results

### 3.1. Anthropometric, Endocrine–Metabolic, and Body Composition Variables

Appendix A summarizes the study participants’ clinical, anthropometric, endocrine–metabolic, and body composition data. As expected, the pre-gestational BMI of women delivering LGA babies was significantly higher than that of mothers delivering AGA and SGA newborns [13]. At birth, SGA infants were shorter and leaner than AGA and LGA newborns; at the age of 12 months, BMI Z-scores were comparable between the subgroups, as previously reported [3] (Appendix A). The existing differences in glucose homeostasis parameters and body composition observed at birth or at the age of 4 months between the subgroups disappeared by the age of 12 months.

### 3.2. Non-Targeted Lipidomics-Based Analysis

After an exhaustive analysis, 1221 and 310 features were detected using the LC–MS ESI(+) and LC–MS ESI(−) approaches, respectively. After data normalization, filtration by CVs in QC samples (<20%), and based on the VIP threshold (VIP > 1.0) and *p* ≤ 0.05 in the Mann–Whitney U test, 104 and 55 compounds were found to be statistically significant for comparisons (LGA vs. AGA; SGA vs. AGA; LGA vs. SGA) by LC–MS ESI(+) and LC–MS ESI(−) analysis, respectively. The fold change for each compound was also evaluated (Appendix A). All metabolites depicting a corrected Mann–Whitney U test *p*-value < 0.05 in at least one of the comparisons are displayed in Table 1. Those compounds with a *p*-value slightly higher than 0.05 (between 0.05 and 0.10) were retained to enhance the biological interpretation.

The PLS-DA models showed a clear discrimination between the three subgroups, displaying good-quality parameters (explained variance, R^2^ ≥ 0.6; predicted variance, Q^2^ ≥ 0.4), with the differences among them lower than 0.3 [29] (Appendix A). Then, the OPLS-DA plots were used to shed light on the most affected lipids that can be used to determine the principal lipid networks altered depending on BW. The quality of the models, together with the outstanding cross-validation results and *p*-values CV-ANOVA of each model, is described in Appendix A.

### 3.3. Lipid Species Modulated in SGA, AGA, and LGA Infants

Several lipid classes—including eicosanoids, oxo fatty acids, glycerophospholipids, glycerolipids, sphingolipids, and sterols—were significantly different between the subgroups (Table 1). Figure 1 provides the LINEX visualization of the lipidomic changes detected. Red nodes represent lipids with increased levels, while blue nodes represent lipids with decreased levels within each comparison (1A, SGA vs. AGA; 1B, LGA vs. AGA; 1C, LGA vs. SGA).

Figure 1A shows that the levels of triglyceride (TG) species were upregulated overall in SGA neonates as compared to AGA infants. The PUFA-enriched TGs 20:4/16:0/O-18:0, TG 16:0/16:0/20:4, TG 14:0/18:3/22:3, and TG 16:0/18:1/22:6, involved in cholesterol efflux, were the most increased species, while diacylglycerol (DG) levels were also markedly upregulated. Conversely, SGA infants showed downregulation of sphingomyelin (SM) species, which are a class of lipids playing a role in in the regulation of transmembrane signaling. Three cholesteryl ester (CE) lipid species were significantly downregulated, while the cholesteryl 11-hydroperoxy-eicosatetraenoate, involved in cholesterol uptake, was markedly increased. Moreover, SGA infants expressed lower levels of lipid species involved in inflammation, such as lysophosphatidylcholine (LPC) and glycerophosphatidylcholine (PC), while other ox-PC lipid species were found to be upregulated. SGA infants had also higher levels of multiple oxylipins, and of two acylcarnitines (i.e., decanoylcarnitine and decenoylcarnitine), essential for fatty acid oxidation. Finally, ascorbyl palmitate was only detected in the SGA subgroup.

Figure 1B depicts the comparisons between the LGA and AGA subgroups. Only minor differences in TG levels were detected, and the same scenario was observed regarding PCs, where only PC 18:2/22:2 was significantly decreased. Unlike SGA infants, LGA infants displayed increased SM, and acylcarnitines (i.e., oleoylcarnitine and palmitoylcarnitine) were also upregulated. Some important oxylipins involved in the formation of eicosanoids showed a distinct pattern in LGA newborns (1(3)-glyceryl-6-keto-PGF1α/2-glyceryl-6-keto-PGF1α, 12,20-DiHETE, 19-hydroxy-PGE2, 5S-HpEPE, 6-hydroxypentadecanedioic acid, and methyl-FA 18:3; 2OOH were undetectable, while 12-HETE levels were extremely low). Cholesteryl 11-hydroperoxy-eicosatetraenoate was also downregulated, as were the levels of several fatty acids (FAs). Finally, 9-HODE, with a potential role in cardiometabolism, was remarkably upregulated.

Figure 1C depicts the differences between LGA and SGA infants, revealing an upregulation of LPCs and PCs in the former—especially LPC 20:3/0:0, PC 16:0/20:3, PC 22:2/16:1, and PC 18:0/20:3. LGA infants also displayed an upregulation of SM levels, lower levels of the glycerolipids DG and TG, and higher oleoylcarnitine and palmitoylcarnitine concentrations. We also detected significantly decreased levels of several FFAs in LGA newborns, including linoleic acid, α-linolenic acid, docosatetraenoic acid, docosahexaenoic acid, and docosapentaenoic acid. Oxylipins displayed the most significant modulation; most of them were only detected in SGA infants, or were drastically downregulated in LGA infants (Table 1).

The cross-validated OPLS-DA model generated (R^2^ = 0.995, Q^2^ = 0.91, *p* CV-ANOVA = 2.02 × 10^−14^), including LC–MS ESI(+) and ESI(−) data as an overview model, demonstrated the remarkable separation of both the SGA and LGA subgroups from the AGA subgroup; this separation was also detected between the SGA and LGA subgroups (Figure 2).

### 3.4. Correlations

Variable influence on projection (VIP) thresholds were used to select the target lipid species potentially related to clinical outcomes. Table 1 depicts the lipids involved in the correlation analysis. Table 2 shows that the anthropometric measures at birth and at 4 months of age were inversely correlated with several lipid species belonging to different families, including eicosanoids, oxo fatty acids, and sterols. In addition, those lipids showed a positive correlation with HOMA-IR at 4 months of age.

## 4. Discussion

In this pilot study, we report for the first time the presence of distinct cord blood lipidome patterns in SGA, LGA, and AGA newborns. Moreover, we describe several associations between target lipid species distinctly modulated in SGA, AGA, and LGA newborns, along with parameters related to growth and glucose homeostasis.

Lipid peroxidation is related to oxidative stress; the imbalance between pro-oxidants and antioxidants leads to overproduction of reactive oxygen species (ROS) [39]. This oxidative-stress-induced damage can play a crucial role in adverse pregnancy outcomes such as fetal growth restriction, resulting in SGA. Although the increase in oxidative stress during pregnancy may be considered a physiological event in normal birth [40], our results reveal an excessive activation of this process in umbilical cord samples from SGA infants as a result of the upregulation of multiple prostaglandins (PGs), eicosanoids, and oxy-PUFAs. Interestingly, excessive PGs and thromboxane have been associated with fetal growth retardation [41]. On the other hand, the lipid 6-keto-PGF1α—a stable degradation product of PGI2 or prostacyclin, and a valuable marker of PGI2 in humans [42,43]—was lower in LGA newborns as compared to SGA and AGA infants. Since PGI2 is a pro-adipogenic molecule, the decreased levels of its precursor in LGA newborns may be reminiscent of a downregulation indicating the presence of sufficient subcutaneous adipose tissue. Moreover, the levels of this marker were inversely correlated with anthropometric measures at birth; most associations were maintained at 4 months of age, and were positively correlated with HOMA-IR, suggesting that the PGI2 precursor may play a role in the fat accretion that occurs during the catch-up process, and in the subsequent changes in insulin sensitivity. The arachidonic acid metabolite 12,20-DiHETE was also differentially modulated in SGA, LGA, and AGA newborns. Lipids derived from arachidonic acids have been implicated in the development of obesity-associated complications, including diabetes and insulin resistance [44,45,46]. In our study, 12,20-DiHETE levels in cord blood were low in LGA newborns and high in SGA infants as compared to AGA newborns, were inversely correlated with weight at birth, and positively correlated with HOMA-IR levels at 4 months of age. Thus, it is tempting to speculate that the higher levels observed in SGA newborns could play a role in the development of the lipotoxicity that follows an excessive catch-up in weight in those infants.

PC is the most abundant phospholipid in mammalians, comprising 40–50% of total cellular glycerophospholipids [46]. PCs play a critical role in membrane-mediated cell signaling, and are an essential source for the formation of lipid mediators, including LPCs. LPC receptors are present in a broad range of tissues and cell types, suggesting critical roles in processes such as reproduction and cardiovascular/neurodegenerative diseases [47]. We found a downregulation of LPC levels and multiple PC species in SGA infants, as compared to AGA infants. Conversely, significantly higher LPC and PC lipid levels were found in LGA vs. SGA infants. Some of these lipid species were positively correlated with anthropometric variables and measures of insulin resistance at birth. Our results are consistent with previous data depicting a positive association between fetal LPC 16:1 and size at birth [48,49]. Lower LPC levels in SGA infants may increase their susceptibility to infection in the long term, since these lipid mediators protect against infection by acting as potent chemoattractants for monocytes, T cells, and natural killer cells, as well as potentiating the activation of T lymphocytes [50]

Fatty acid β-oxidation generates energy from fat, and L-carnitine and acylcarnitines facilitate this process in the mitochondria. Increased production of acylcarnitines has been reported in patients with obesity, insulin resistance, and type 2 diabetes [51]. The pathogenesis of insulin resistance and diabetes has been associated with reduced fatty acid β-oxidation inducing lipotoxicity and impaired insulin signaling [52]. We found significantly higher levels of medium-chain acylcarnitines in SGA infants, while long-chain acylcarnitine levels were higher in LGA vs. AGA infants, together with lower levels of palmitic and oleic acids. The positive correlation of acylcarnitines with BW, and the inverse correlation with insulin sensitivity at 12 months of age, could be related to the surplus of fat present in LGA newborns. Our results are consistent with those of Beken et al. [53], who reported a similar acylcarnitine trend in umbilical cord blood samples for LGA, SGA, and AGA neonates.

The association between TG and cardiovascular risk is well established [54]. Our results show a general increase in PUFA-enriched TG levels in SGA infants, consistent with previous studies showing increased TG levels in SGA newborns [55,56,57]. Other studies related high TG concentrations in sera to the increased hepatic secretion of TG-rich lipoproteins and decreased lipolysis or reuptake of TG-rich lipoproteins [56]. Accordingly, SGA newborns may have a limited capacity to store TG in adipocytes, in addition to a potential increase in de novo biosynthesis.

Limitations of this study include the relatively low number of patients (only 12 individuals per group) and the lack of information about maternal lifestyles (no data regarding dietary habits, exercise, or sleep quality). In addition, the availability of lipidomic data at 4 and 12 months of age would have been potentially useful to identify the specific lipid species modulated during the postnatal period and conceivably involved in metabolic adaptation. The major strength of this study is the longitudinal follow-up up to the age of 12 months, including clinical, endocrine–metabolic, and body composition assessments.

## 5. Conclusions

In conclusion, the distinct lipidome patterns observed in SGA, AGA, and LGA newborns may play a role in adipose tissue remodeling and future metabolic risks. Maternal dietary interventions focused on these lipid species may potentially provide long-term benefits for the metabolic health of the offspring.

## Figures and Tables

**Figure 1 nutrients-14-03760-f001:**
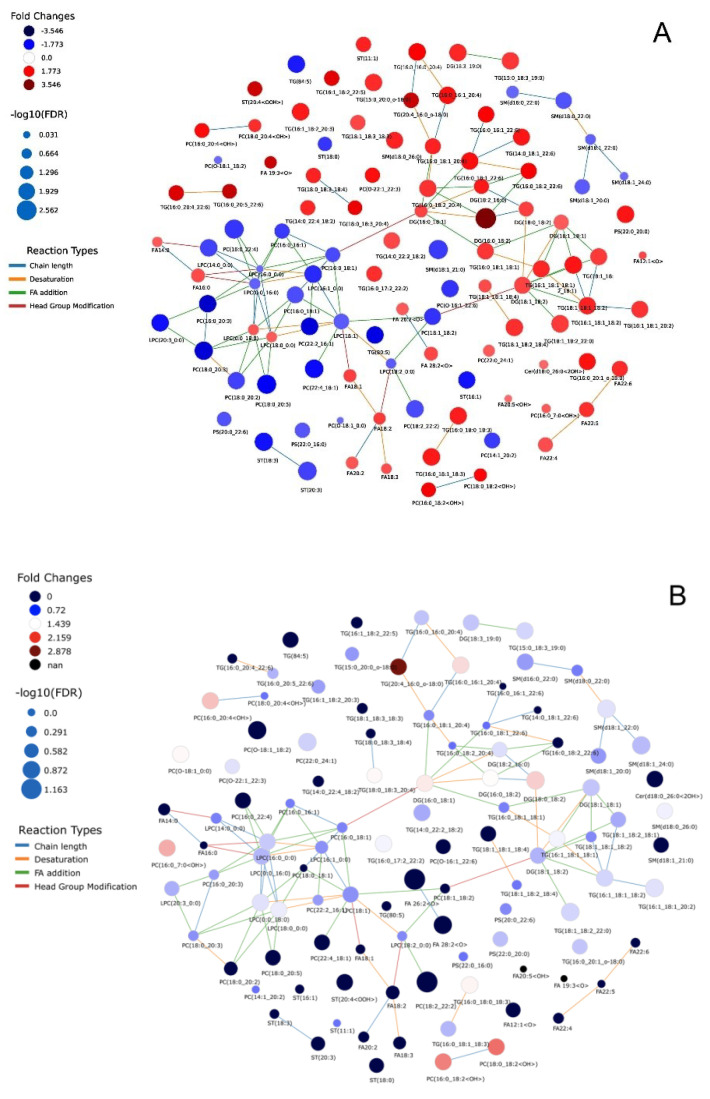
LINEX Lipid network of the umbilical cord plasma samples. Plots presents node size scaled by −log10 of *p*-values and coloured by fold change for the comparison between (**A**) SGA vs. AGA, (**B**) LGA vs. AGA, (**C**) LGA vs. SGA. Blue colours indicate lower levels of lipids in the (**A**) SGA, (**B**,**C**) LGA group compared to (**A**,**B**) AGA and (**C**) SGA samples. All edges are coloured by reaction type.

**Figure 2 nutrients-14-03760-f002:**
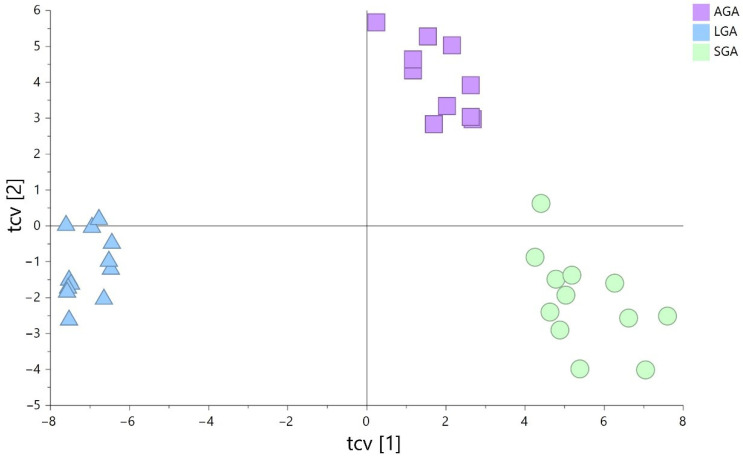
Cross-validated OPLS-DA model including LC–MS ESI(+) and ESI(−) data. Pareto log scaling was used for the model (green circles: small-for-gestational-age (SGA) samples; blue triangles: large-for-GA (LGA) samples; purple squares: appropriate-for-GA (AGA) samples.

**Table 1 nutrients-14-03760-t001:** Main compounds found in umbilical cord plasma samples from small- (SGA, *n* = 12), appropriate- (AGA, *n* = 12) and large-for-gestational-age (LGA, *n* = 12) newborns, depicting statistical significance for the comparisons among study subgroups. *p*-Values marked with “*” correspond to the corrected *p*-Value obtained after Benjamini-Hochberg (BH) correction test.

							**LGA vs. AGA**			**SGA vs. AGA**			**LGA vs. SGA**		
**Name**	** *m/z* **	**RT (min)**	**Formula**	**CV (%)**	** *p* ** **-Value**	** *p* ** **-Value BH**	**FC**	** *p-* ** **Value**	**VIP**	**FC**	** *p-* ** **Value**	**VIP**	**FC**	** *p-* ** **Value**	**VIP**
Fatty Acyls															
1(3)-Glyceryl-6-keto-PGF1α/ 2-glyceryl-6-keto-PGF1α	443.2649	0.84	C_23_H_40_O_8_	2.8	4.9 × 10^−6^	1.4 × 10^−4^	−LGA	-	4.40	5.41	0.20	1.73	−LGA	-	4.07
11-HEDE	301.1990	0.89	C_20_H_36_O_3_	8.6	0.0051	0.051	0.35	0.77	0.61	2.99	0.045 *	2.04	0.12	0.0029 *	1.75
12,20-DiHETE	355.2276	2.52	C_20_H_32_O_4_	3.9	6.9 × 10^−6^	1.4 × 10^−4^	−LGA	-	4.61	1.59	0.83	1.25	−LGA	-	3.97
12-HETE	339.2329	3.21	C_20_H_32_O_3_	1.1	0.0017	0.018	0.01	0.011	2.47	0.87	0.97	1.36	0.01	0.0043 *	2.50
19-Hydroxy-PGE2	429.2497	0.82	C_20_H_34_O_6_	5.4	2.2 × 10^−7^	1.4 × 10^−5^	−LGA	-	3.41	22.20	0.0025 *	3.45	−LGA	-	3.84
3-Hexenyl 3-hydroxybutanoate; WE 10:1; O	245.1397	0.78	C_10_H_18_O_3_	8.1	1.8 × 10^−7^	1.4 × 10^−5^	-	-	-	+SGA	-	4.80	−LGA	-	3.39
5S-HpEPE	353.2122	1.71	C_20_H_30_O_4_	2.8	6.9 × 10^−6^	1.4 × 10^−4^	−LGA	-	4.76	0.99	0.67	0.22	−LGA	-	4.15
6-Hydroxypentadecanedioic acid	287.1866	0.84	C_15_H_28_O_5_	6.1	5.5 × 10^−6^	1.4 × 10^−4^	−LGA	-	3.93	4.83	0.28	1.73	−LGA	-	3.66
9-HODE	297.2421	2.88	C_18_H_32_O_3_	7.1	0.0033	0.021	3.55	0.045	1.56	1.98	0.013 *	0.96	1.79	0.62	1.11
Ascorbyl palmitate	473.2750	0.85	C_22_H_38_O_7_	4.1	1.8 × 10^−7^	1.4 × 10^−5^	-	-	-	+SGA	-	5.49	−LGA	-	3.80
Eicosadienoic acid	307.264	3.84	C_20_H_36_O_2_	2.4	0.053	0.23	0.91	0.37	0.16	1.18	0.14	0.67	0.77	0.022 *	0.56
FA 26:2; O	467.3739	2.72	C_26_H_48_O_3_	2.3	5.2 × 10^−4^	0.0069	0.69	0.048 *	0.90	1.17	0.13	0.55	0.58	0.0030 *	0.90
FA 28:2; O	495.4052	3.78	C_28_H_52_O_3_	4.2	0.0012	0.014	0.70	0.048 *	0.82	1.28	0.093	0.80	0.55	0.0071 *	0.97
FAHFA(18:1-(10-O-16:0)	595.4939	3.80	C_34_H_64_O_4_	1.4	0.0014	0.016	0.71	0.048 *	0.75	1.33	0.082	0.83	0.54	0.0077 *	0.94
FAHFA(30:1)	539.4310	2.95	C_30_H_56_O_4_	2.6	1.1 × 10^−4^	0.0017	0.65	0.037 *	0.92	1.36	0.041	0.92	0.47	0.00038 *	1.10
Hexacosanedioic acid; FA26:1;O2	425.3636	3.52	C_26_H_50_O_4_	2.4	9.4 × 10^−5^	0.0016	0.67	0.037 *	0.96	1.36	0.021	0.91	0.49	0.0027 *	1.07
Methyl-FA 18:3;2OOH	415.2363	0.81	C_19_H_32_O_6_	5.1	1.3 × 10^−6^	6.7 × 10^−5^	−LGA	-	3.74	8.42	0.045 *	2.52	−LGA	-	3.82
Decanoylcarnitine	316.2482	0.92	C_17_H_33_NO_4_	7.9	0.045	0.067	1.30	0.027	0.38	1.50	0.033	1.12	0.87	0.36	0.15
Decenoylcarnitine	314.2326	0.86	C_17_H_31_NO_4_	8.6	0.034	0.061	1.31	0.039	0.38	1.49	0.045 *	1.12	0.88	0.58	0.09
Oleoylcarnitine	426.3576	2.43	C_25_H_47_NO_4_	5.5	0.0091	0.043	1.46	0.0035	0.65	1.15	0.22	0.49	1.27	0.038 *	0.40
Palmitoylcarnitine	400.3420	2.25	C_23_H_45_NO_4_	5.4	0.0017	0.018	1.46	0.029 *	0.64	1.18	0.14	0.48	1.24	0.029	0.38
α-Linolenic acid	277.217	2.55	C_18_H_30_O_2_	2.4	0.080	0.26	0.86	0.34	0.28	1.25	0.16	0.67	0.68	0.045	0.64
Docosatetraenoic acid	331.264	3.65	C_22_H_36_O_2_	1.4	0.013	0.094	0.91	0.18	0.30	1.18	0.093	0.65	0.76	0.0194 *	0.56
Docosahexaenoic acid; DHA	327.2329	2.77	C_22_H_32_O_2_	2.1	0.0057	0.054	0.97	0.83	0.14	1.55	0.045 *	1.07	0.63	0.020 *	0.75
Docosapentaenoic acid; DPA	329.249	3.16	C_22_H_34_O_2_	5.8	0.030	0.16	1.03	0.76	0.13	1.56	0.019	0.99	0.66	0.026 *	0.61
Eicosadienoic acid	307.264	3.84	C_20_H_36_O_2_	2.4	0.053	0.23	0.91	0.37	0.16	1.18	0.14	0.67	0.77	0.022 *	0.56
Hexacosanedioic acid	425.364	3.52	C_26_H_50_O_4_	2.4	9.4 × 10^−5^	0.0016	0.67	0.037 *	0.96	1.36	0.021	0.91	0.49	0.0027 *	1.07
Linoleic acid	279.233	3.05	C_18_H_32_O_2_	1.5	0.037	0.19	0.80	0.37	0.32	1.40	0.081	0.87	0.57	0.022 *	0.81
Myristic acid	227.202	2.56	C_14_H_28_O_2_	3.2	0.13	0.36	0.91	0.45	0.16	1.22	0.21	0.68	0.75	0.054 *	0.57
Oleic acid	281.249	3.63	C_18_H_34_O_2_	1.1	0.058	0.23	0.95	1.00	0.05	1.41	0.070	0.94	0.68	0.032 *	0.59
Palmitic acid	255.233	3.46	C_16_H_32_O_2_	2.4	0.11	0.30	0.98	0.85	0.10	1.26	0.074	0.74	0.78	0.045	0.48
Glycerolipids															
DG 16:0/18:2/0:0	610.5393	11.34	C_37_H_68_O_5_	6.0	1.5 × 10^−5^	0.0015	1.45	0.053	0.70	3.55	0.011 *	3.46	0.41	0.0083 *	1.33
TG 14:0/18:3/22:3	917.6996	13.57	C_57_H_98_O_6_	6.7	0.023	0.057	0.86	0.18	0.27	1.80	0.061	1.46	0.48	0.055 *	1.00
TG 14:0/22:2/18:2	921.7310	13.94	C_57_H_102_O_6_	4.8	0.024	0.057	1.20	0.078	0.31	1.29	0.021 *	0.71	0.93	0.62	0.29
TG 15:0/18:3/19:0	895.7153	13.94	C_55_H_100_O_6_	6.0	0.016	0.049	1.32	0.056	0.48	1.41	0.020 *	0.93	0.94	0.65	0.40
TG 15:0/20:0/O-18:0	901.8020	13.55	C_57_H_98_O_6_	14.7	0.017	0.049	1.15	0.21	0.27	1.48	0.019 *	1.05	0.78	0.068	0.42
TG 16:0/16:0/20:4	872.7700	13.55	C_55_H_98_O_6_	4.3	0.017	0.049	1.27	0.21	0.42	1.80	0.019 *	1.54	0.70	0.095	0.66
TG 16:0/18:0/18:3	879.7411	13.94	C_55_H_100_O_6_	3.8	0.014	0.047	1.47	0.054	0.59	1.56	0.020 *	1.18	0.94	0.59	0.50
TG 16:0/18:1/18:1	876.8000	13.56	C_55_H_102_O_6_	8.8	0.030	0.059	1.08	0.48	0.19	1.40	0.026 *	0.95	0.77	0.052	0.45
TG 16:0/18:1/18:3	877.7253	13.55	C_55_H_98_O_6_	5.1	0.014	0.047	1.23	0.14	0.36	1.58	0.019 *	1.22	0.78	0.15	0.52
TG 16:0/18:1/20:4	898.7850	13.56	C_57_H_100_O_6_	16.5	0.055	0.073	1.10	0.47	0.25	1.53	0.035 *	1.21	0.72	0.052	0.56
TG 16:0/18:1/22:6	922.7880	13.56	C_59_H_100_O_6_	7.3	0.021	0.055	1.02	0.87	0.19	1.69	0.020 *	1.38	0.60	0.055 *	0.78
TG 16:0/18:2/20:4	896.7700	13.24	C_57_H_98_O_6_	7.5	0.027	0.058	1.01	0.93	0.08	1.42	0.04 *	0.99	0.71	0.055 *	0.51
TG 16:0/18:2/22:6	920.7690	13.23	C_59_H_98_O_6_	2.6	0.025	0.057	0.91	0.43	0.40	1.88	0.037 *	1.59	0.48	0.013	1.08
TG 16:1/18:1/18:1	874.7860	13.94	C_55_H_100_O_6_	2.8	0.024	0.057	1.41	0.081	0.57	1.52	0.024 *	1.12	0.93	0.57	0.49
TG 16:1/18:1/18:2	872.7641	13.54	C_55_H_98_O_6_	6.9	0.0098	0.044	1.33	0.035	0.38	1.53	0.020 *	1.03	0.87	0.16	0.46
TG 16:1/18:1/20:2	905.7567	13.94	C_57_H_102_O_6_	7.0	0.014	0.047	1.35	0.030	0.46	1.42	0.021 *	1.02	0.95	0.71	0.36
TG 16:1/18:2/20:3	896.7702	13.56	C_57_H_98_O_6_	15.7	0.014	0.047	1.16	0.22	0.28	1.60	0.019 *	1.24	0.73	0.051	0.59
TG 18:1/18:1/18:2	900.7971	13.56	C_57_H_102_O_6_	13.8	0.011	0.044	1.09	0.41	0.19	1.61	0.013 *	1.30	0.68	0.0082	0.60
TG 18:1/18:2/22:0	958.8783	13.94	C_61_H_112_O_6_	12.7	0.0064	0.035	1.32	0.013	0.44	1.38	0.019 *	0.87	0.95	0.67	0.31
TG 20:4/16:0/O-18:0	907.7515	12.44	C_57_H_104_O_5_	4.6	0.0011	0.014	2.88	0.0017	1.45	2.50	0.014 *	2.47	1.15	0.89	0.90
TG 84:5	1367.23	14.69	C_87_H_164_NO_9_	7.7	0.022	0.056	0.73	0.085	0.80	0.63	0.018 *	1.23	1.15	0.46	0.31
Glycerophospholipids															
LPC 14:0/0:0	468.3082	1.68	C_22_H_46_NO_7_P	5.6	0.0046	0.028	1.07	0.43	0.13	0.76	0.014 *	0.90	1.41	0.025 *	0.53
LPC 16:1/0:0	494.3239	1.84	C_24_H_48_NO_7_P	5.7	0.0025	0.019	1.12	0.27	0.23	0.68	0.013 *	1.35	1.65	0.010 *	0.81
LPC 18:1/0:0	580.3615	2.70	C_26_H_52_NO_7_P	2.9	0.0050	0.051	1.11	0.11	0.37	0.86	0.031	0.70	1.29	0.011 *	0.59
LPC 20:3/0:0	546.3549	2.57	C_28_H_52_NO_7_P	6.2	2.2 × 10^−4^	0.0045	1.21	0.11	0.31	0.59	0.018 *	1.65	2.04	0.0064 *	1.07
PC 16:0/16:1	732.5533	7.14	C_40_H_78_NO_8_P	1.6	0.010	0.044	1.06	0.67	0.22	0.73	0.025 *	1.13	1.45	0.020	0.56
PC 16:0/18:1	760.5852	8.86	C_42_H_82_NO_8_P	1.0	0.0069	0.035	1.09	0.68	0.21	0.80	0.020 *	0.79	1.37	0.012	0.44
PC 16:0/20:3	784.5846	8.55	C_44_H_82_NO_8_P	1.9	1.7 × 10^−4^	0.0045	1.15	0.26	0.29	0.44	0.013 *	2.46	2.64	0.0064 *	1.38
PC 16:0/22:4	810.6010	8.28	C_46_H_84_NO_8_P	2.9	0.062	0.076	1.10	0.52	0.28	0.77	0.045 *	0.77	1.43	0.078	0.46
PC 16:0/22:4	810.6008	8.66	C_46_H_84_NO_8_P	2.8	0.0019	0.018	0.85	0.069	0.30	0.71	0.011 *	1.02	1.20	0.039	0.25
PC 18:0/20:2	814.6328	11.43	C_46_H_88_NO_8_P	3.9	0.013	0.047	0.89	0.22	0.25	0.77	0.018 *	0.78	1.16	0.12	0.21
PC 18:0/20:3	870.6224	11.29	C_46_H_86_NO_8_P	2.8	1.0 × 10^−4^	0.0016	1.03	0.92	0.20	0.42	0.0012 *	1.73	2.44	0.0016 *	1.19
PC 18:0/20:4; 12OH	826.5950	5.99	C_46_H_84_NO_9_P	12.6	0.13	0.14	1.03	0.91	1.51	1.46	0.042 *	1.52	0.71	0.29	1.39
PC 18:0/20:5	808.5845	8.06	C_46_H_82_NO_8_P	2.0	3.0 × 10^−4^	0.0051	0.81	0.13	0.33	0.54	0.012 *	1.81	1.49	0.0083 *	0.57
PC 18:2/22:2	838.6314	11.06	C_48_H_88_NO_8_P	5.7	0.0013	0.015	0.74	0.040 *	0.47	0.79	0.018 *	0.74	0.95	0.45	0.12
PC 22:2/16:1	812.6160	10.96	C_46_H_86_NO_8_P	4.2	1.5 × 10^−4^	0.0045	1.15	0.34	0.27	0.47	0.0033 *	2.18	2.45	0.0064 *	1.25
PC 22:4/18:1	836.6157	10.25	C_48_H_86_NO_8_P	2.7	0.0024	0.019	0.82	0.092	0.30	0.59	0.0033 *	1.61	1.39	0.028 *	0.51
PC O-18:1/0:0	508.3760	3.05	C_26_H_54_NO_6_P	5.0	0.030	0.059	1.45	0.030	0.60	0.99	0.61	0.43	1.47	0.026	0.63
PC O-18:1/18:2	770.6050	9.31	C_44_H_84_NO_7_P	16.1	0.027	0.058	0.82	0.089	0.43	0.70	0.019 *	0.96	1.18	0.71	0.19
PS 20:0/22:6	864.5696	8.86	C_48_H_82_NO_10_P	18.7	0.016	0.049	1.09	0.34	0.16	0.83	0.048 *	0.60	1.32	0.011	0.41
PS 22:0/20:0	893.7000	13.55	C_48_H_94_NO_10_P	6.5	0.019	0.053	1.26	0.18	0.39	1.63	0.019 *	1.30	0.77	0.15	0.57
Sphingolipids															
SM 16:0; O2/22:0	761.653	11.51	C_43_H_89_N_2_O_6_P	4.8	0.026	0.058	1.15	0.060	0.19	0.93	0.13	0.27	1.24	0.055 *	0.30
SM 18:0; O2/22:0	789.6857	11.94	C_45_H_93_N_2_O_6_P	6.2	0.0033	0.021	1.07	0.26	0.11	0.84	0.026 *	0.52	1.27	0.028 *	0.36
SM 18:0; O2/26:0	845.7410	13.17	C_49_H_101_N_2_O_6_P	12.0	0.037	0.061	1.40	0.098	0.49	1.54	0.020 *	1.29	0.90	0.31	0.47
SM 18:1; O2/20:0	819.6590	11.64	C_43_H_89_N_2_O_6_P	5.3	0.018	0.11	1.11	0.21	0.37	0.85	0.016	0.69	1.31	0.020 *	0.60
SM 18:1; O2/21:0	811.6042	8.66	C_44_H_89_N_2_O_6_P	3.0	0.0026	0.019	0.87	0.11	0.27	0.73	0.011 *	0.95	1.20	0.042	0.25
SM 18:1; O2/22:0	787.6677	11.54	C_45_H_91_N_2_O_6_P	7.0	0.0070	0.035	1.36	0.018	0.43	0.92	0.29	0.31	1.47	0.046 *	0.52
SM 18:1; O2/24:0	815.7000	11.93	C_47_H_95_N_2_O_6_P	2.2	0.014	0.047	1.24	0.015	0.31	0.97	0.61	0.19	1.28	0.0070	0.35
Sterol Lipids															
22:0-Glc-Cholesterol	871.7420	13.25	C_55_H_98_O_7_	2.4	0.030	0.059	1.10	0.67	0.21	1.45	0.020 *	1.03	0.76	0.078	0.49
3-Deoxyvitamin D3	369.3510	12.52	C_27_H_44_	4.9	0.047	0.067	1.16	0.49	1.58	1.88	0.030 *	2.02	0.62	0.048	1.71
CE 16:1	640.6020	14.57	C_43_H_74_O_2_	3.6	0.020	0.054	0.93	0.68	0.42	0.61	0.0137 *	1.76	1.53	0.079	0.60
CE 18:3	664.6019	14.11	C_45_H_74_O_2_	22.0	0.013	0.047	0.89	0.18	0.57	0.61	0.013 *	1.61	1.46	0.40	0.53
CE 20:3	713.5632	14.69	C_47_H_78_O_2_	2.9	0.0066	0.035	0.89	0.19	0.24	0.77	0.013 *	0.73	1.16	0.079	0.22
Cholesteryl 11-hydroperoxy-eicosatetraenoate	743.5375	12.41	C_47_H_76_O_4_	3.2	1.0 × 10^−4^	0.0045	0.52	0.049	1.95	2.46	0.026 *	2.72	0.21	0.0038 *	1.97

**Table 2 nutrients-14-03760-t002:** Correlations between cord blood lipids and infant data at birth, at 4 months and at 12 months and the respective changes (0–4 and 0–12 months).

	**1(3)-Glyceryl-6-keto-PGF1alpha/** **2-glyceryl-6-keto-PGF1α**		**11-HEDE**		**12-HETE**		**12,20-DiHETE**		**19R-hydroxy-PGE1**		**5S-HpEPE**		**6-Hydroxypentadecanedioic acid**		**Methyl-FA 18:3;2OOH**	
** *At birth* **	R	*p*	R	*p*	R	*p*	R	*p*	R	*p*	R	*p*	R	*p*	R	*p*
Weight	−0.82	**<0.0001**	0.36	0.0279	−0.56	**0.0003**	−0.80	**<0.0001**	−0.91	**<0.0001**	−0.83	**<0.0001**	−0.82	**<0.0001**	−0.83	**<0.0001**
Length	−0.73	**<0.0001**	0.40	0.0144	−0.46	0.0042	−0.70	**<0.0001**	−0.84	**<0.0001**	−0.74	**<0.0001**	−0.73	**<0.0001**	−0.74	**<0.0001**
BMI Z-score	−0.75	**<0.0001**	0.39	0.0178	−0.54	**0.0007**	−0.72	0.0035	−0.85	**<0.0001**	−0.75	**<0.0001**	−0.75	**<0.0001**	−0.77	**<0.0001**
HOMA-IR	0.13	0.2034	0.49	0.0029	−0.02	0.5755	0.08	0.6980	0.06	0.6797	0.09	0.5482	0.13	0.2323	0.11	0.4670
** *At 4 months* **																
Weight	−0.59	**0.0003**	0.45	0.0087	−0.23	0.2027	−0.57	**0.0006**	−0.70	**<0.0001**	−0.62	**0.0001**	−0.61	**0.0002**	−0.62	**0.0001**
Length	−0.61	**0.0002**	0.28	0.1087	−0.24	0.1761	−0.56	**0.0008**	−0.67	**<0.0001**	−0.63	**0.0001**	−0.63	**<0.0001**	−0.60	**0.0002**
BMI Z-score	−0.33	0.0642	0.57	**0.0008**	−0.09	0.2725	−0.35	0.0520	−0.46	0.0088	−0.37	0.0368	−0.35	0.0543	−0.39	0.0287
HOMA-IR	0.44	0.0033	0.13	0.3602	0.18	0.3165	0.37	0.0521	0.48	0.0026	0.42	0.0492	0.52	**<0.0001**	0.46	0.0040
** *At 12 months* **																
Weight	−0.38	0.0209	0.37	0.0251	−0.26	0.1290	−0.44	0.0070	−0.46	0.0053	−0.43	0.0086	−0.34	0.0408	−0.40	0.0171
Length	−0.33	0.0464	0.17	0.3152	−0.33	0.0520	−0.40	0.0146	−0.38	0.0212	−0.35	0.0376	−0.24	0.1559	−0.37	0.0257
BMI Z-score	−0.22	0.1995	0.43	0.0088	−0.04	0.8369	−0.24	0.1708	−0.30	0.0817	−0.28	0.1017	−0.26	0.1275	−0.22	0.2020
HOMA-IR	0.30	0.2893	−0.10	0.3981	0.23	0.4132	0.26	0.3879	0.24	0.3960	0.32	0.1219	0.32	0.1039	0.28	0.1461
** Δ *0–4 months* **																
Weight	0.05	0.7797	0.24	0.1723	0.28	0.1104	0.06	0.7188	−0.00	0.9765	0.01	0.9170	0.01	0.9171	0.02	0.9133
Length	0.08	0.6620	−0.15	0.3884	0.22	0.2218	0.11	0.5377	0.14	0.4316	0.06	0.7043	0.05	0.7556	0.10	0.5647
BMI Z-score	0.66	**<0.0001**	−0.12	0.5141	0.53	0.0018	0.63	**0.0001**	0.70	**<0.0001**	0.65	**<0.0001**	0.65	**<0.0001**	0.64	**<0.0001**
HOMA-IR	0.22	0.4699	−0.24	0.4134	0.10	0.7117	0.18	0.5190	0.30	0.1867	0.22	0.3641	0.29	0.0911	0.25	0.2475
** Δ *0–12 months* **																
Weight	0.11	0.5274	0.20	0.2305	0.07	0.6765	0.01	0.9130	0.08	0.6267	0.05	0.7468	0.16	0.3583	0.10	0.5524
Length	0.32	0.0553	−0.18	0.2762	0.08	0.6426	0.23	0.1781	0.37	0.0245	0.32	0.0561	0.42	0.0108	0.30	0.0768
BMI Z-score	0.60	**0.0002**	−0.12	0.6655	0.51	0.0020	0.56	**0.0006**	0.65	**<0.0001**	0.56	**0.0005**	0.57	**0.0004**	0.62	**<0.0001**
HOMA-IR	0.07	0.7758	−0.20	0.0698	0.16	0.4879	0.06	0.8720	0.08	0.8618	0.11	0.2786	0.09	0.7417	0.07	0.7753

BMI, body mass index; HOMA-IR, homeostasis model assessment insulin resistance. Bold means *p* < 0.001. Δ means modulation (increment or decrement).

## Data Availability

The data that support the findings of this study are available from the corresponding authors upon request.

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
