# Peer review of "Birth Weight and Early Postnatal Outcomes: Association with the Cord Blood Lipidome"

_nutrients, 2022, doi:10.3390/nu14183760_

Round 1

Reviewer 1 Report

This is my review on the article “Birthweight & Early Postnatal Outcomes: Association with Cord Blood Lipidome”. This is a well written manuscript. The aim was to investigate the association among birth weight, early postnatal outcomes and cord blood serum lipidome. This is an important study as adult metabolic risks may originate from gestational and newborn weight imbalance. 

The longitudinal follow-up up to age 12 months, including clinical, endocrine-metabolic and body composition assessments is indeed remarkable. 

Limitations have been mentioned by authors (are indeed the low number of patients). However, this study presents significant results regarding the lipidome pattern observed in SGA, AGA and LGA newborns. These results may be useful in future studies regarding maternal innervations.  

Author Response

We thank the Reviewer for the positive evaluation of our manuscript and appreciate the constructive comments. Changes to the English language have been incorporated, as suggested. The Introduction section has also been extended to incorporate more information and new references, as requested (lines 45-66, tracked version).

Reviewer 2 Report

There are few things to edit; however, the most common

error that I saw was comma placement and run-on sentences.

Line 50: Lipids are a highly diverse molecule class and are important players in multiple

physiologic functions, including energy storage, and membrane architecture or signalling.

Edit: molecule → molecular

signalling → signaling

Line 57: Moreover, dynamic shifts in the maternal lipidome across pregnancy, including transfer

of lipids containing polyunsaturated fatty acids (PUFA) from maternal to fetal circulation have

been demonstrated, suggesting that maternal lipids may modulate cord blood lipid levels

influencing growth.

Edit: Moreover, dynamic shifts in the maternal lipidome across pregnancy, including

transfer of lipids containing polyunsaturated fatty acids (PUFA) from maternal to fetal

circulation, have been demonstrated, suggesting that maternal lipids may modulate cord blood

lipid levels influencing growth.

Line 60: In addition, the role of n-6 and n-3 fatty acids and their metabolism into biologically

active eicosanoids is still a milestone in the field since eicosanoids may play a key role in

inflammation].

Edit: In addition, the role of n-6 and n-3 fatty acids and their metabolism into biologically

active eicosanoids is still a milestone in the field, since eicosanoids may play a key role

in inflammation.

Line 103: Maternal age at delivery, smoking habits, parity, pre-gestational weight, and

weight and body mass index (BMI) at delivery [the latter defined as weight (in kg)/square

of height (in meters)], were obtained from the hospital clinical records.

Edit: Maternal age at delivery, smoking habits, parity, pre-gestational weight, and

weight and body mass index (BMI) at delivery [the latter defined as weight (in kg)/square

of height (in meters)] were obtained from the hospital clinical records.

Line 172: The neutral loss (NL) of water were also considered for both ion modes.

Edit: The neutral loss (NL) of water was also considered for both ion modes.

Line 296: Unlike SGA infants, LGA displayed increased SM and tacylcarnitines (oleoylcarnitine

and palmitoylcarnitine) were also found to be upregulated.

Edit: Unlike SGA infants, LGA displayed increased SM, and tacylcarnitines

(oleoylcarnitine and palmitoylcarnitine) were also found to be upregulated.

Author Response

Comments and Suggestions for Authors

There are few things to edit; however, the most common error that I saw was comma placement and run-on sentences.

We thank the Reviewer for this observation. We have now (hopefully) corrected these mistakes in the revised version.

Line 50: Lipids are a highly diverse molecule class and are important players in multiple physiologic functions, including energy storage, and membrane architecture or signalling.

Edit: molecule → molecular

signalling → signaling

OK, modified as suggested.

Line 57: Moreover, dynamic shifts in the maternal lipidome across pregnancy, including transfer of lipids containing polyunsaturated fatty acids (PUFA) from maternal to fetal circulation have been demonstrated, suggesting that maternal lipids may modulate cord blood lipid levels influencing growth.

Edit: Moreover, dynamic shifts in the maternal lipidome across pregnancy, including transfer of lipids containing polyunsaturated fatty acids (PUFA) from maternal to fetal circulation, have been demonstrated, suggesting that maternal lipids may modulate cord blood lipid levels influencing growth.

 OK, modified as suggested.

Line 60: In addition, the role of n-6 and n-3 fatty acids and their metabolism into biologically active eicosanoids is still a milestone in the field since eicosanoids may play a key role in inflammation].

Edit: In addition, the role of n-6 and n-3 fatty acids and their metabolism into biologically active eicosanoids is still a milestone in the field, since eicosanoids may play a key role in inflammation.

OK, modified as suggested.

Line 103: Maternal age at delivery, smoking habits, parity, pre-gestational weight, and weight and body mass index (BMI) at delivery [the latter defined as weight (in kg)/square of height (in meters)], were obtained from the hospital clinical records.

Edit: Maternal age at delivery, smoking habits, parity, pre-gestational weight, and weight and body mass index (BMI) at delivery [the latter defined as weight (in kg)/square of height (in meters)] were obtained from the hospital clinical records.

OK, modified as suggested.

Line 172: The neutral loss (NL) of water were also considered for both ion modes.

Edit: The neutral loss (NL) of water was also considered for both ion modes.

OK, modified as suggested.

Line 296: Unlike SGA infants, LGA displayed increased SM and tacylcarnitines (oleoylcarnitine and palmitoylcarnitine) were also found to be upregulated.

Edit: Unlike SGA infants, LGA displayed increased SM, and tacylcarnitines (oleoylcarnitine and palmitoylcarnitine) were also found to be upregulated.

OK, modified as suggested.

Reviewer 3 Report

The authors investigate the link between birthweight,   Postnatal Outcomes and the Cord Blood Lipidome 

Abstract

The content is clear, but lacking in the specific findings of the study which should be included

The introduction would be benefited from a hypothesis

Materials and methods

Is the study sufficiently powered?

Were women excluded if they are taking medication?

The limitations were not sufficiently described, and they need more clarification

Author Response

Comments and Suggestions for Authors

The authors investigate the link between birthweight,   Postnatal Outcomes and the Cord Blood Lipidome

Abstract

The content is clear, but lacking in the specific findings of the study which should be included.

We thank the Reviewer for his/her comments.

Following the Reviewer’s suggestions, we have now modified the Abstract to include a summary of the requested specific findings (lines 27-30, tracked version). As the Reviewer knows, there is a limit of 200 words for this section that makes it difficult to extend further the information.

The introduction would be benefited from a hypothesis

We agree with this comment and have now added a sentence in the Introduction (lines 70-73, tracked version) describing the hypothesis of the study.

Materials and methods

Is the study sufficiently powered?

This is a very good point. We agree that studying 12 patients per group allows drawing limited conclusions regarding the effect/role of lipid species in the metabolic status of AGA, SGA and AGA individuals. Here, we were indeed limited by the amount of sample available at birth from those individuals who had longitudinal data. We have now added the word “pilot” both in the Abstract (line 22, tracked version) and in the first sentence of the Discussion (line 455, tracked version) to clarify that this is a (novel) first step that needs confirmation and that nevertheless may open a new window in the field of maternal dietary interventions that modulate offspring metabolic status in patients with metabolic risk. We already clarified in the limitations paragraph (lines 558-563, tracked version) that the low number of subjects was indeed a limitation; now we have even specified those numbers.

Were women excluded if they are taking medication?

Yes, women were excluded if taking medications. To avoid further confusion, we have now specifically listed this exclusion criterion (lines 113-114, tracked version).

The limitations were not sufficiently described, and they need more clarification

We thank the Reviewer for this comment. The limitations paragraph has been extended as requested (lines 558-563, tracked version).